# Scalable Equilibrium Propagation via Intermediate Error Signals for Deep Convolutional CRNNs

**Jiaqi Lin**                                                            *jkl6467@psu.edu*
*School of Electrical Engineering and Computer Science*
*The Pennsylvania State University, University Park, PA 16802, USA*

**Malyaban Bal**                                                         *mjb7906@psu.edu*
*School of Electrical Engineering and Computer Science*
*The Pennsylvania State University, University Park, PA 16802, USA*

**Abhronil Sengupta**                                                    *sengupta@psu.edu*
*School of Electrical Engineering and Computer Science*
*The Pennsylvania State University, University Park, PA 16802, USA*

**Reviewed on OpenReview:** *https://openreview.net/forum?id=iXFmzKpPNA*

## Abstract

Equilibrium Propagation (EP) is a biologically inspired local learning rule first proposed for convergent recurrent neural networks (CRNNs), in which synaptic updates depend only on neuron states from two distinct phases. EP estimates gradients that closely align with those computed by Backpropagation Through Time (BPTT) while significantly reducing computational demands, positioning it as a potential candidate for on-chip training in neuromorphic architectures. However, prior studies on EP have been constrained to shallow architectures, as deeper networks suffer from the vanishing gradient problem, leading to convergence difficulties in both energy minimization and gradient computation. To alleviate the vanishing gradient problem in deep EP networks, we propose a novel EP framework that incorporates layer-wise learning signals to provide auxiliary supervision, which enhances the convergence of neuron dynamics. This is the first work to integrate knowledge distillation and local error signals into EP, enabling the training of significantly deeper architectures. Our proposed approach achieves state-of-the-art performance on the CIFAR-10 and CIFAR-100 datasets, showcasing its scalability on deep VGG architectures. These results represent a significant advancement in the scalability of EP, suggesting that intermediate learning signals can extend the practical applicability of EP to deeper architectures. The project repository is available at `https://github.com/NeuroCompLab-psu/ScalableEPInterErr`.

## 1 Introduction

Recent advances in Equilibrium Propagation (EP) have garnered significant interest as a biologically plausible method for training energy-based models, particularly convergent recurrent neural networks (CRNNs) with bidirectional neurons (Scellier & Bengio, 2017). These bidirectional connections form a computational circuit that facilitates the exchange of data and error signals across layers, enabling neurons to evolve toward stable states that minimize an energy function. In contrast to EP, Backpropagation (BP) separates the forward and backward computational processes. The forward pass integrates weighted inputs at each neuron, while the backward pass explicitly computes gradients based on error signals. This aspect is often considered to be biologically implausible (Crick, 1989). EP, in comparison, enables error signal propagation through backward connections. Small perturbations applied at the output layer influence preceding layers (Figure 1 (a)). EP updates weights by evaluating the difference between neuron states before and after the application of error signals, resembling Spike-Timing-Dependent Plasticity (STDP) (Bi & Poo, 1998; Scellier & Bengio, 2017).

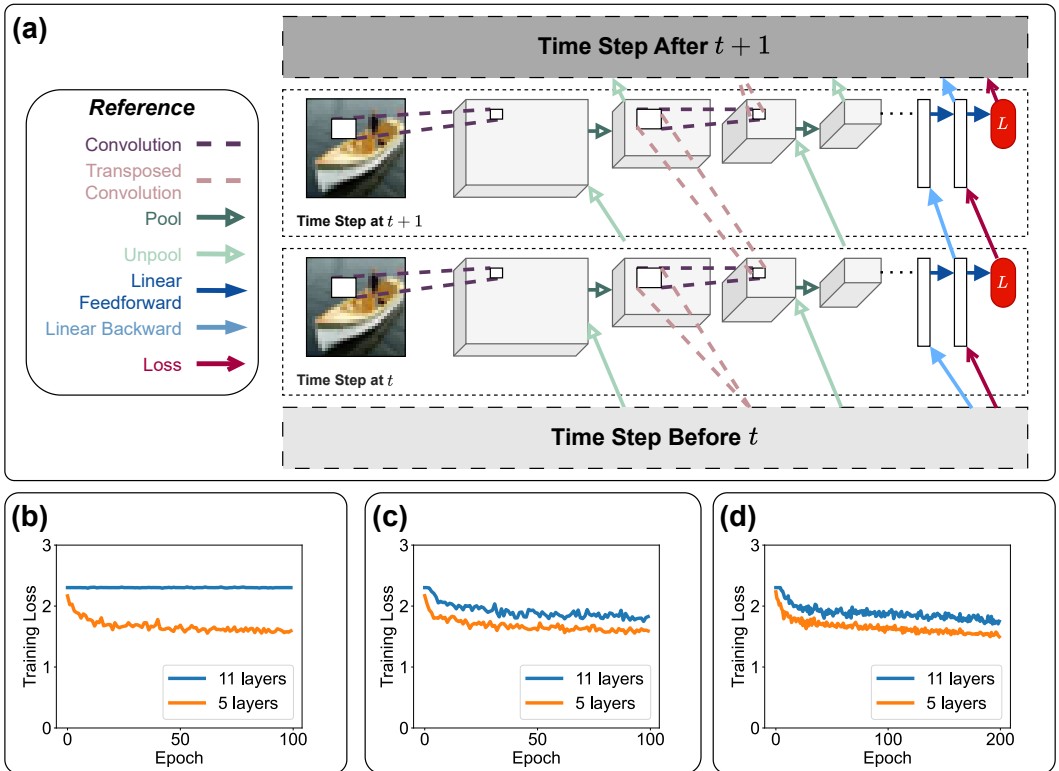

Figure 1: (a) Overview of convolutional CRNNs trained via the EP framework, showing information flow during forward and backward passes at time steps $t$ and $t + 1$. Forward passes apply convolutional and pooling operations, while backward passes use transposed convolutions and unpooling. In linear layers, forward and backward passes use matrix multiplication, with weights transposed during the backward pass. The loss at the output layer propagates backward through transposed weights. (b) Training loss on the CIFAR-10 dataset using VGG-5 and VGG-11 CRNNs trained with unaugmented EP. Deeper networks show higher loss due to vanishing gradients. (c) Training loss on the CIFAR-10 dataset for VGG-5 and VGG-11 CRNNs trained with EP augmented by local error signals. (d) Training loss on the CIFAR-10 dataset for VGG-5 and VGG-11 CRNNs trained with EP augmented by knowledge distillation.

This update mechanism exhibits both spatial and temporal locality, making it well-suited for neuromorphic computing (Martin et al., 2021). Additionally, theoretical analyses of EP (Scellier & Bengio, 2019) have shown it to be a close approximation of Backpropagation Through Time (BPTT) (Almeida, 1990; Pineda, 1987), highlighting its promise as a biologically plausible alternative to BPTT.

Early EP frameworks were limited to shallow fully connected networks due to first-order bias in gradient estimates from finite nudging (Ernoult et al., 2019; 2020; O'Connor et al., 2019; Scellier & Bengio, 2017). To address this, later works introduced both positive and negative nudge phases to cancel the bias in nudge terms and improve gradient estimation (Laborieux et al., 2021). Recent advances (Bal & Sengupta, 2023; Laborieux et al., 2021; Lin et al., 2024) have further narrowed the performance gap between EP and BP across vision and language tasks.

However, existing EP frameworks face significant scalability challenges. As neural networks deepen, convergence issues emerge, leading to a notable decline in performance (Figure 1). This degradation arises from the vanishing gradient problem (Hochreiter, 1998), initially identified in recurrent neural networks (RNNs) due to long-term temporal dependencies. Similarly, EP-trained deep CRNNs encounter extended temporal dependencies, where both forward-propagated inputs and backward error signals attenuate across layers dur-

ing simulation. Consequently, intermediate neuron states fail to retain essential information, disrupting the transmission of input and error signals. This effect prevents neuron dynamics from converging, and gradient computations (which heavily depend on neuron state changes during the nudge phase) collapse entirely. To tackle this issue in BPTT-trained RNNs, prior works have proposed various architectural and algorithmic modifications:

**Activation functions.** Activation functions like Rectified Linear Unit (ReLU) (Nair & Hinton, 2010) and Gaussian Error Linear Unit (GELU) (Hendrycks & Gimpel, 2016) help mitigate vanishing gradients in BP-trained networks by maintaining gradients for positive inputs. In EP, the Hard Sigmoid (Courbariaux et al., 2015) is currently utilized, which clamps inputs beyond thresholds, leading to loss of information. Intuitively, substituting the Hard Sigmoid function with either ReLU or GELU is expected to mitigate the vanishing gradient problem. Nonetheless, the implementation of these activation functions does not evidently resolve the convergence issue.

**Initialization strategy.** Weight initialization strategies (Glorot & Bengio, 2010; He et al., 2015) and neuron state initialization techniques (Laborieux et al., 2021) have been employed to promote effective signal propagation by maintaining variance across layers, thereby preventing gradients from vanishing or exploding. However, in the context of the EP framework, this approach has proven to be insufficient for addressing scalability challenges (Gammell et al., 2021).

**Batch normalization.** Internal covariate shift, where the input distribution to each layer changes across epochs, has been shown to hinder the training process (Ioffe, 2015). While batch normalization effectively mitigates this issue, it lacks biological plausibility and has not been integrated into the EP framework.

**Skip connections.** Skip connections, used in architectures like ResNet (He et al., 2016), DenseNet (Huang et al., 2017), U-Net (Ronneberger et al., 2015), and Transformers (Vaswani, 2017), have been explored to address the vanishing gradient problem. Within the EP framework, inspired by the principles of small-world topology (Watts & Strogatz, 1998), the concept of skip connections has been examined to address the vanishing gradient problem (Gammell et al., 2021). This is achieved by replacing standard connections with randomly assigned layer-skipping connections. Similarly, Liu & Chen (2025) integrated EP with residual connections to alleviate vanishing gradients. However, these approaches are primarily limited to linear layers and have not been evaluated on more complex benchmarks such as CIFAR-10 and CIFAR-100.

In this work, we propose a novel framework to address the vanishing gradient problem in EP by introducing additional learning signals that enhance information flow, thereby bridging the performance gap between shallow and deep network architectures (Figure 1). The primary contributions of this work are as follows:

- We present **a scalable EP framework** that, for the first time, enables the successful training of **deep CRNNs —surpassing the limitations of prior EP studies**, which were restricted to five-layer convolutional architectures.

- We provide **empirical evidence of vanishing gradients in deep EP networks**, highlighting how convergence issues occur during both the minimization of the scalar primitive function and the gradient computation process.

- We enhance EP training through the introduction of intermediate error signals. To the best of our knowledge, this is the **first integration of knowledge distillation and local error signals within the EP framework**. Our method **achieves state-of-the-art performance** on CIFAR-10 and CIFAR-100 datasets using deep VGG architectures.

## 2 EP Foundations

### 2.1 Convolutional CRNNs

EP was originally utilized for training CRNNs (Scellier & Bengio, 2017). CRNNs are characterized by receiving a static input $x$ and having recurrent dynamics that converge to a steady state. Ernoult et al. (2019) and Laborieux et al. (2021) introduced the scalar primitive function $\Phi(x, \xi^t, w)$ to describe the dynamics of

convolutional CRNNs and successfully trained convolutional CRNNs using EP on the MNIST dataset. The scalar primitive function is defined as:

$$\Phi(x, \xi^t, w) = \sum_{i=0}^{N_c-1} \left( \xi_{i+1}^t \circ \mathcal{P}(w_i * \xi_i^t) \right) + \sum_{i=N_c}^{N_t-1} \left( {\xi_{i+1}^t}^\mathsf{T} \cdot w_i \cdot \xi_i^t \right) \tag{1}$$

Here, $N_\mathrm{t}$ represents the total number of layers consisting of $N_\mathrm{c}$ convolutional layers and $N_\mathrm{l}$ linear layers. The pooling function is denoted as $\mathcal{P}(\cdot)$. The parameters for layer $i$ are denoted as $w_i$, and $\xi^t$ are the neuron states at time $t$. Distinct operations in convolutional and linear layers are represented as follows: $*$ signifies the convolution operation, $\circ$ indicates the generalized scalar product between two vectors of the same dimensionality, and $\cdot$ represents the linear mapping operation (formal definitions of these operations are provided in Appendix F). The neuron state dynamics are then formulated as $\xi^{t+1} = \frac{\partial \Phi(x, \xi^t, w)}{\partial \xi}$.

## 2.2 Equilibrium Propagation

The training procedure of EP, composed of two distinct phases, is adapted to perform gradient descent in CRNNs on a loss function $L(\xi_\mathrm{out}^t, \hat{y})$ defined between the target $\hat{y}$ and the output activations $\xi_\mathrm{out}^t$ (Scellier & Bengio, 2017). In the first (free) phase, a static input $x$ is presented and the network evolves over $T_\mathrm{free}$ time steps until convergence to a stable state $\xi^*$ that serves as a fixed point of the gradient field of the primitive function $\Phi(x, \xi^t, w)$. In the second (nudge) phase, a nudge term $\beta \frac{\partial L(\xi_\mathrm{out}^t, \hat{y})}{\partial \xi}$ is introduced to the output layer $\xi_\mathrm{out}^t$, where $\beta$ denotes the nudging factor. The resulting neuron dynamics for the output layer is given by:

$$\xi^{t+1} = \frac{\partial \Phi(x, \xi^t, w)}{\partial \xi} - \beta \frac{\partial L(\xi_\mathrm{out}^t, \hat{y})}{\partial \xi} \tag{2}$$

The nudging term shifts the saturated states of the neural network towards another stable state $\xi^\beta$ that is closer to the true label in $T_\mathrm{nudge}$ time steps.

**Theorem 2.1** (Theorem 2 of Scellier & Bengio (2019)). *The gradient of the objective function $L$ with respect to $w$ can be estimated by computing the divergence of the two stable states:*

$$\frac{\partial L}{\partial w} = \lim_{\beta \to 0} \frac{1}{\beta} \left( \frac{\partial \Phi(x, \xi^\beta, w)}{\partial w} - \frac{\partial \Phi(x, \xi^*, w)}{\partial w} \right) \tag{3}$$

To address the estimation bias introduced by a positive nudging factor $\beta$—which can lead to inaccurate gradient estimates (Laborieux et al., 2021)—a third phase is incorporated into the training process. This additional phase applies a nudging factor of $-\beta$ to counteract the bias. In this work, we adopt this three-phase training procedure. Let $\xi^{-\beta}$ represent the network's steady state at the end of the third phase. Under this formulation, Equation 3 becomes:

$$\frac{\partial L}{\partial w} = \lim_{\beta \to 0} \frac{1}{2\beta} \left( \frac{\partial \Phi(x, \xi^\beta, w)}{\partial w} - \frac{\partial \Phi(x, \xi^{-\beta}, w)}{\partial w} \right) \tag{4}$$

In this formulation, EP exploits spatial and temporal locality to update the weights between connected layers.

## 2.3 Equivalence of EP and BPTT

BPTT can also be applied to train CRNNs. At each iteration of the free phase, the network runs for $T_\mathrm{free}$ time steps until it saturates to a stable state $\xi^*$. BPTT operates by unfolding the neuron states $\xi$ across time, allowing the model to capture temporal dependencies in the input data. The loss function $L(\xi_\mathrm{out}^{T_\mathrm{free}}, \hat{y})$ is evaluated at the final time step $T_\mathrm{free}$, and gradients are computed by backpropagating the error signals through time. The weight update rule for BPTT is expressed as:

$$\frac{\partial L}{\partial w} = \sum_{t=1}^{T_\mathrm{free}} \frac{\partial L}{\partial \xi^t} \frac{\partial \xi^t}{\partial w} \tag{5}$$

**Theorem 2.2.** *Suppose that the convergence of neuron states $\xi$ reached its steady state $\xi^*$ within $T_{\text{free}}$ time steps in the first phase, the gradient updates computed by the EP algorithm $\nabla^{\text{EP}}$ (Equation 3 and Equation 4) approximate the gradients computed by the BPTT algorithm $\nabla^{\text{BPTT}}$ (Equation 5) with an infinitesimal $\beta$ in the first $T_{\text{nudge}}$ time steps of the second phase:*

$$\nabla^{\text{EP}}(\beta, w) \xrightarrow[\beta \to 0]{} -\nabla^{\text{BPTT}}(\beta, w) \tag{6}$$

Theoretical analyses supporting Theorem 2.2 have been provided in Ernoult et al. (2019); Scellier & Bengio (2019). Notably, this equivalence holds for both fully connected and convolutional architectures, including networks with pooling operations.

## 3 EP Training Augmentation

### 3.1 EP Framework Suffers From the Vanishing Gradient Problem

In this section, we demonstrate that the existing EP framework exhibits the vanishing gradient problem as the depth of CRNN architectures increases. This issue was originally identified in the optimization of RNNs, where BP-based methods struggle to capture long-term temporal dependencies (Hochreiter, 1998). In the EP framework, weight updates rely entirely on the equilibrium states of neurons during both the free and nudge phases. However, in deep CRNNs, backward-propagated error signals diminish across layers, failing to sufficiently perturb the equilibrium states. This results in the cancellation of neuron state differences used for weight updates, ultimately leading to vanishing gradients. As illustrated in Appendix A, neuron states in the traditional EP framework progressively diminish over training epochs. Consequently, information becomes concentrated in the initial and final layers, while intermediate neurons receive minimal gradient signals.

### 3.2 Mitigate the Vanishing Gradient Problem in EP Using Intermediate Learning Signals

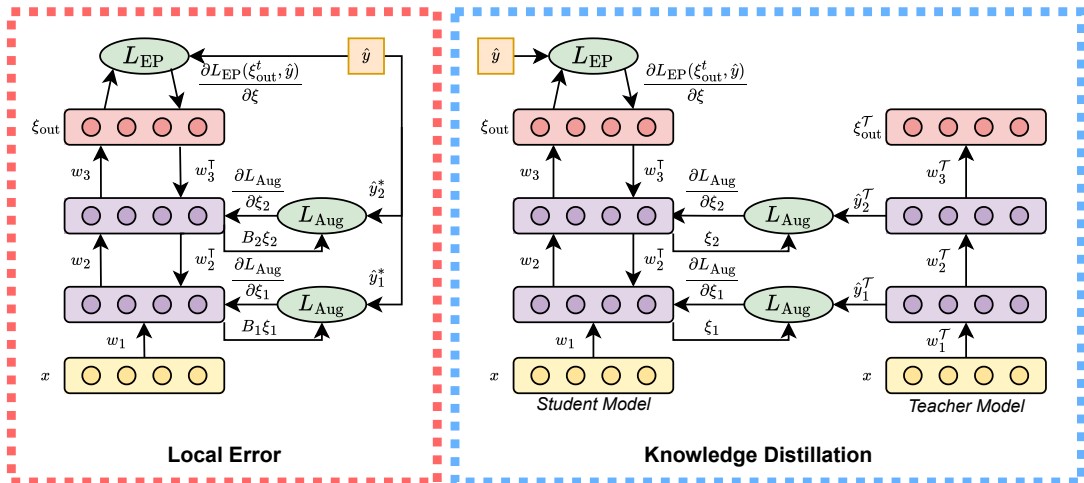

Figure 2: Simplified architecture of augmented EP framework. (Left) Local error method augments the intermediate representations by incorporating error signals computed at each layer using intermediate read-outs and pseudo-targets. (Right) Knowledge distillation enhances neural dynamics by aligning intermediate representations between the student and teacher models. Here, $\xi$ denotes neuron states, $w$ are the weights, $x$ is the input, $\hat{y}$ are the true labels, and $\hat{y}^*$ are the pseudo-targets. $B$ represents the projection matrices, $\hat{y}^{\mathcal{T}}$ corresponds to the teacher model's output logits, and $L_{\text{Aug}}$ and $L_{\text{EP}}$ denote the intermediate and output losses, respectively. Both augmentation methods show improved performance in our experiments.

To mitigate the vanishing gradient problem in EP, we introduce intermediate learning signals at select layers (Figure 2), thereby enabling the training of deeper networks. This study explores two types of signals:

**Local Error (LE).** In the local error settings (Frenkel et al., 2021), where pseudo-targets $\hat{y}_i^*$ provide guidance for each layer (which corresponds to $\hat{y}$ in our experiments), intermediate readouts are calculated as $\hat{\xi}_i^t = B_i \xi_i^t$ for layers $i \in \Upsilon$ (set of layers with intermediate learning signals), where $B_i$ is a randomly initialized matrix associated with layer $i$ that projects the neuron states $\xi_i^t$ onto pseudo-targets $\hat{y}_i^*$. The corresponding loss function is defined as:

$$L_{\text{Aug}}(\xi_i^t, \hat{y}_i^*) = Loss(\hat{\xi}_i^t / \tau, \hat{y} / \tau) \tag{7}$$

Here, $\tau$ denotes the temperature parameter used to smoothen both the intermediate readout and pseudo-target distributions to facilitate learning (Hinton et al., 2015), and *Loss* is the loss function. The weight update rules for the projection matrices $B_i$, derived under the EP framework, are provided in Section 4.3.

**Knowledge Distillation (KD).** Following the approach of Hinton et al. (2015), KD incorporates learning signals from a separate, pre-trained teacher model on the same task. For layers $i \in \Upsilon$, the loss is computed between intermediate layer outputs (logits) of the student and teacher models:

$$L_{\text{Aug}}(\xi_i^t, \hat{y}_i^*) = Loss(\xi_i^t / \tau, \hat{y}_i^{\mathcal{T}} / \tau) \tag{8}$$

Here, $\xi_i^t$ and $\hat{y}_i^{\mathcal{T}}$ denote the logits from the student and teacher models respectively at layer $i$ and time step $t$. In this setting, the pseudo-targets $\hat{y}_i^*$ correspond to the teacher logits $\hat{y}_i^{\mathcal{T}}$.

Let $L_{\text{Aug}}$ denote the loss associated with intermediate learning signals, scaled by a factor $\kappa$, and $L_{\text{EP}}$ measure the discrepancy between the network's output and the true label. The modified loss function for the augmented EP framework is defined as follows:

$$L_{\text{Total}} = L_{\text{EP}}(\xi_{\text{out}}^t, \hat{y}) + \kappa \sum_{i \in \Upsilon} L_{\text{Aug}}(\xi_i^t, \hat{y}_i^*) \tag{9}$$

**Theorem 3.1** (Convergence of the Augmented EP Framework). *Let $\Phi_{\text{Aug}}(x, \xi^t, w)$ be the scalar primitive function of the augmented EP framework in the nudge phase defined by:*

$$\begin{aligned}
\Phi_{\text{Aug}}(x, \xi^t, w) &= \Phi(x, \xi^t, w) - \beta L_{\text{Total}} \\
&= \Phi(x, \xi^t, w) - \beta L_{\text{EP}}(\xi_{\text{out}}^t, \hat{y}) \\
&\quad - \beta \kappa \sum_{i \in \Upsilon} L_{\text{Aug}}(\xi_i^t, \hat{y}_i^*)
\end{aligned} \tag{10}$$

*Here, $\Phi(x, \xi^t, w)$ is the standard scalar primitive function defined in Equation 1, and $\Upsilon$ is a set of layers designated for additional signaling. Assume that: (1) $\frac{\partial \Phi(x, \xi^t, w)}{\partial \xi}$ is Lipschitz continuous with constant $K_1 < 1$; (2) the gradients of the loss functions, $\frac{\partial L_{\text{EP}}}{\partial \xi}$ and $\frac{\partial L_{\text{Aug}}}{\partial \xi}$, are Lipschitz continuous with constants $K_2$ and $K_3$, respectively; and (3) the hyperparameters $\beta$ and $\kappa$ are chosen such that $K_1 + |\beta|K_2 + |\beta\kappa|K_3 < 1$. Under these conditions, the discrete-time neuron dynamics, $\xi^{t+1} = \frac{\partial \Phi_{\text{Aug}}(x, \xi^t, w)}{\partial \xi}$, converge to a fixed point as $t \to \infty$.*

The proposed EP framework retains convergence guarantees toward a local minimum of the scalar primitive function, consistent with the discrete-time setting described in Ernoult et al. (2019). A theoretical proof of Theorem 3.1 under discrete-time dynamics is provided in Appendix G, which outlines a sufficient condition for convergence in such settings. In a three-phase training algorithm, Theorem 3.1 is applied to both nudge phases to guarantee the convergence of neuron states to $\xi_{\text{Aug}}^{\beta}$ and $\xi_{\text{Aug}}^{-\beta}$, respectively. Empirically, we demonstrate that the augmented EP framework converges to stable states (Figure 3) and effectively mitigates the vanishing gradient problem (Appendix A). Importantly, the framework preserves biological plausibility by relying solely on spatially and temporally local information for weight updates.

## 3.3 Gradient Estimation with Additional Learning Signal

Laborieux et al. (2021) demonstrated that the gradient estimate produced by the three-phase EP algorithm (Equation 4) has an error bound of $O(\beta^2)$, derived via a second-order Taylor expansion of $\frac{\partial \Phi(x, \xi^\beta, w)}{\partial w}$ and

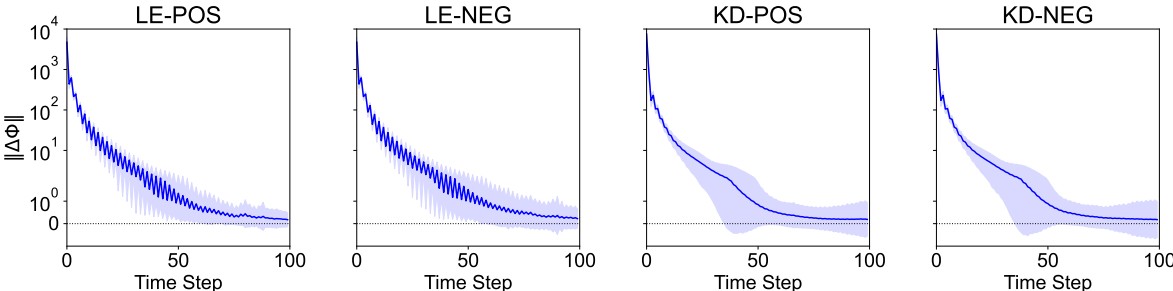

Figure 3: The gradient of VGG-7 scalar primitive function, trained by the augmented EP framework, converges to steady states in the nudge phase (data shown for the 25th epoch). Results are based on 100 random CIFAR-10 samples. The blue line shows the mean energy, and the shaded area indicates the standard deviation. LE-POS and LE-NEG denote the augmented EP framework with LE signals applied during the nudge phase using $\beta$ and $-\beta$, respectively. Similarly, KD-POS and KD-NEG refer to the augmented EP framework with KD signals for $\beta$ and $-\beta$ nudge phases, respectively.

$\frac{\partial \Phi(x, \xi^{-\beta}, w)}{\partial w}$ (where $\Phi$ includes the loss function in the nudge phase). The resulting gradient $\nabla_{\text{3phase}}^{\text{EP}}(\beta, w)$ can be interpreted as the average of two gradients from the two-phase algorithm (Theorem 2.1), $\nabla_{\text{2phase}}^{\text{EP}}(\beta, w)$ and $\nabla_{\text{2phase}}^{\text{EP}}(-\beta, w)$, which are computed using converged neuron states with $\beta > 0$ and $\beta < 0$, respectively:

$$\nabla_{\text{3phase}}^{\text{EP}}(\beta, w) = \frac{\nabla_{\text{2phase}}^{\text{EP}}(\beta, w) + \nabla_{\text{2phase}}^{\text{EP}}(-\beta, w)}{2} = -\frac{\partial L_{\text{EP}}(\xi_{\text{out}}^t, \hat{y})}{\partial w} + O(\beta^2) \tag{11}$$

The augmented EP framework extends the three-phase algorithm (Equation 11) by incorporating additional learning signals, $\beta \kappa \sum_{i \in \Upsilon} L_{\text{Aug}}(\xi_i^t, \hat{y}_i^*)$, into the primitive function $\Phi(x, \xi^t, w)$. Theorem 3.2 provides the corresponding gradient estimation for this augmented formulation.

**Theorem 3.2.** *With $\beta \to 0$, the gradient estimation of the proposed augmented EP framework $\nabla_{\text{Aug}}^{\text{EP}}(\beta, w)$ follows:*

$$\nabla_{\text{Aug}}^{\text{EP}}(\beta, w) = -\kappa \sum_{i \in \Upsilon} \left( \frac{\partial L_{\text{Aug}}(\xi_i^t, \hat{y}_i^*)}{\partial w} \right) - \frac{\partial L_{\text{EP}}(\xi_{\text{out}}^t, \hat{y})}{\partial w} + O(\beta^2) \tag{12}$$

The theoretical justification for Theorem 3.2 follows the same gradient derivation as presented in Scellier & Bengio (2019); Laborieux et al. (2021) by modifying the loss function.

## 4 Results

### 4.1 Experimental Setup

To assess the effectiveness of the proposed EP framework described in Section 3, we evaluate the performance of deep convolutional CRNNs on the CIFAR-10 and CIFAR-100 datasets (Krizhevsky et al., 2009), with results summarized in Table 1. Dataset details are provided in Appendix B. All experiments are conducted in Python using the PyTorch library (Paszke, 2019), on an NVIDIA RTX A5000 GPU with 24 GB of memory. Network weights are initialized using the uniform Kaiming initialization scheme (He et al., 2015), and optimal hyperparameters (detailed in Appendix D) are applied to maximize performance. The datasets are normalized and augmented with random cropping and horizontal flipping. Stochastic Gradient Descent (SGD) with momentum and weight decay is used for training, combined with Cross-Entropy Loss (CE) at the output layer, Soft Target Loss (Hinton et al., 2015) at layers with intermediate loss signals, and the learning rate scheduling strategy introduced by Loshchilov & Hutter (2016).

### 4.2 Performance

**State-of-the-art Performance.** In this section, we demonstrate that the augmented EP framework achieves state-of-the-art performance on the CIFAR-10 dataset (Table 1a). The evaluated CRNN archi-

Table 1: Training and testing accuracy (%) of CRNNs trained using the EP framework on CIFAR-10 and CIFAR-100 datasets. Results are averaged over 5 independent runs, with standard deviation less than 0.4. LE denotes local error augmentation, and KD denotes knowledge distillation. The VGG-5 network used in prior works consists of four $3 \times 3$ convolutional layers with 128–256–512–512 feature maps (stride 1), followed by a $2 \times 2$ max pooling layer (stride 2).

| Model | Train | Test |
|---|---|---|
| VGG-5 (Laborieux et al., 2021) | - | 88.32 |
| VGG-5 (Laydevant et al., 2021) | - | 84.34 |
| VGG-5 (Laborieux & Zenke, 2022) | - | 88.60 |
| VGG-7-LE | 98.70 | 88.99 |
| VGG-7-KD | 97.57 | 89.67 |

(a) Evaluated on the CIFAR-10 dataset.

| Model | Train | Test |
|---|---|---|
| VGG-5 (Laborieux & Zenke, 2022) | - | 61.6 |
| VGG-11-LE | 84.59 | 62.01 |
| VGG-10-KD | 77.09 | 63.68 |
| VGG-13-LE | 75.56 | 62.22 |
| VGG-12-KD | 79.79 | 64.75 |

(b) Evaluated on the CIFAR-100 dataset.

tecture consists of five convolutional layers with 64, 128, 256, 512, and 512 channels, respectively, each using a kernel size of 3, stride 1, and padding 1. This is followed by a fully connected layer with 512 hidden units. Intermediate learning signals are applied to the 3rd and 5th convolutional layers.

**Evaluation on deeper architectures.** To further evaluate scalability, we extend our experiments to the more challenging CIFAR-100 dataset using deeper architectures: VGG-10 (11) and VGG-12 (13) for KD and LE approaches. The number of linear layers is treated as a tunable component, selected based on the best-performing setting for each case. In the LE setting, intermediate learning signals are applied to the last layer of the 2nd through 5th convolutional blocks. The KD setting augments the last layer of every convolutional block with intermediate learning signals. These results confirm that the proposed framework scales effectively without compromising performance. The detailed CRNN configurations are provided in Appendix C. Table 1b reports the performance of CRNNs trained using EP, demonstrating state-of-the-art results on the CIFAR-100 dataset.

**Comparison of Augmentation Methods.** Our empirical findings indicate that KD generally achieves higher accuracy than local error LE augmentation, assuming a well-trained teacher model is available. However, KD typically requires a longer warm-up phase during early training. We compare the convergence behavior of the proposed EP framework when augmented with LE versus KD signals. As shown in Figure 4, LE achieves over 70% training and test accuracy before epoch 15, whereas KD takes more than 45 epochs to reach comparable levels. The loss curves further support this trend. LE results in a much faster reduction in training loss, particularly within the first 50 epochs.

Despite these differences, both augmentation strategies introduce only marginal computational overhead compared to the standard EP framework. LE incurs a small cost due to the inclusion of projection weights, while KD's overhead is related to the complexity of the teacher model. As shown in Table 2, memory usage remains within acceptable increases across all VGG-based CRNN architectures on the CIFAR-10 and CIFAR-100 datasets. Importantly, both methods provide substantial gains in performance and scalability with minimal resource trade-offs in comparison with BPTT.

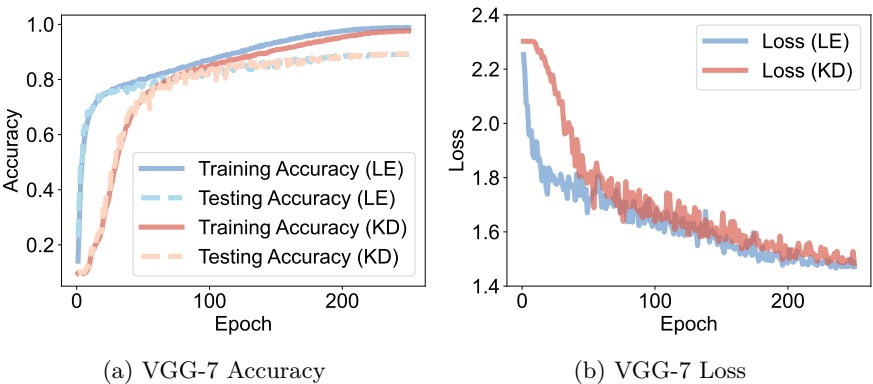

(a) VGG-7 Accuracy

(b) VGG-7 Loss

Figure 4: Performance of VGG-7 network on the CIFAR-10 dataset, trained utilizing proposed EP frameworks. The hyperparameters are reported in Appendix D. LE denotes local error augmentation, and KD denotes knowledge distillation.

### 4.3 Ablation Studies

**Effect of signal magnitude.** Inspired by learning rate scheduling strategies (Paszke, 2019), we propose an epoch-dependent scheduler that dynamically adjusts the magnitude of the intermediate learning signal $\kappa$. In contrast to static signals, which require carefully tuned hyperparameters to approach optimal performance, this adaptive modulation improves CRNNs' performance and broadens the viable hyperparameter range. We evaluate three scheduling variants: linear, exponential, and cosine annealing (Loshchilov & Hutter, 2016). Their respective decay patterns are illustrated in Appendix E.While cosine annealing generally performs well, linear scheduling yields slightly better results in certain knowledge distillation settings, particularly with VGG-12-KD. In contrast, the exponential scheduler decays too quickly, yielding insufficient learning signals and suboptimal performance.

**Weighted knowledge distillation.** One key limitation of our knowledge distillation framework is that it does not include a learnable linear mapping to align the student's logits with those of the teacher. This is essential when the output dimensions or distributions of the two models differ. However, the standard EP framework does not inherently support updating the weights of such mappings. To address this, we introduce an EP-compatible formulation that, conceptually, treats each intermediate KD layer as a separate one-layer sub-network. Each sub-network receives the student's logits as input and produces weighted outputs that are compared to the corresponding teacher logits. The weights of these mappings are updated using the standard EP-based rules (Laborieux et al., 2021) for the penultimate layer. The scalar primitive function describing the optimization of these sub-networks is integrated into the overall scalar primitive function through $L_{\text{Aug}}$, enabling holistic network updates without significant computational overhead. The associated loss function, as defined in Equation 8, is given by:

$$L_{\text{Aug}}(\hat{y}_i^{\text{rd}}, \hat{y}_i^*) = Loss(\hat{y}_i^{\text{rd}}/\tau, \hat{y}_i^{\mathcal{T}}/\tau)$$
$$\text{where} \quad \hat{y}_i^{\text{rd}} = w_i^{\text{map}} \cdot \xi_i^t \tag{13}$$

Here, $w_i^{\text{map}}$ is a linear mapping from student logits $\xi_i^t$ to teacher logits $\hat{y}_i^{\mathcal{T}}$. The proposed design enables gradient-based optimization of the linear weights via standard EP dynamics. The dynamics of neurons within the layers $i \in \Upsilon$ follow $\xi_i^{t+1} = \frac{\partial \Phi_{\text{Aug}}(x, \xi^t, w)}{\partial \xi_i}$ in the nudge phase:

$$\xi_i^{t+1} = \frac{\partial \Phi(x, \xi^t, w)}{\partial \xi_i} - \beta\kappa \frac{\partial L_{\text{Aug}}(\hat{y}_i^{\text{rd}}, \hat{y}_i^*)}{\partial \xi_i} \tag{14}$$

Equation 14 allows neuron states to saturate to $\xi_i^\beta, \xi_i^{-\beta}$ in the positive and negative nudge phases, respectively. Based on the standard EP framework (Laborieux et al., 2021) for the penultimate layer, the update of weights $w_i^{\text{map}}$ for the layers $i \in \Upsilon$ leverage spatial locality in computation by considering saturated neuron

Table 2: GPU memory consumption (in MB) of CRNNs trained using the EP framework on the CIFAR-10 and CIFAR-100 datasets. We compare three training variants: standard EP (STD), EP augmented with local error signals (LE), and EP augmented with knowledge distillation (KD). Results are reported for VGG-7 on CIFAR-10 dataset and VGG-10 (11) and VGG-12 (13) on CIFAR-100 dataset, where the STD and BPTT is evaluated using both VGG-11 and VGG-13 architectures. The batch size is configured to be 128, and $T_{\text{free}}$ is 140 for BPTT.

| Model | VGG-7 | VGG-10 (11) | VGG-12 (13) |
|---|---|---|---|
| Dataset | CIFAR-10 | CIFAR-100 | |
| LE | 791 | 895 | 1351 |
| KD | 855 | 983 | 1431 |
| STD | 711 | 869 | 1285 |
| BPTT* | 8324 $(9.7\times)$ | 13726 $(13.9\times)$ | 25954 $(18.1\times)$ |

* The notation $N\times$ indicates that BPTT consumes $N$ times more GPU memory than EP augmented with KD, which has highest memory usage within EP frameworks.

Table 3: Training and testing accuracy (%) and GPU memory consumption (in MB, with a batch size of 128) of CRNNs trained using the EP framework on the CIFAR-10 dataset. Results are averaged over five independent runs, with standard deviations below 0.3. KDW refers to knowledge distillation with a linear mapping, while KD denotes knowledge distillation without a linear mapping.

| Model | Train | Test | Memory |
|---|---|---|---|
| VGG-7-KD | 97.57 | **89.67** | 855 |
| VGG-7-KDW | 92.81 | 87.93 | 1176 |

states $\xi_i^{\beta}, \xi_i^{-\beta}$ as well as the discrepancy between the weighted logits of the student $\hat{y}_i^{\text{rd},\beta}$, $\hat{y}_i^{\text{rd},-\beta}$ and the logits of teacher $\hat{y}_i^{\mathcal{T}}$:

$$\Delta w_i^{\text{map}} = -\beta\kappa \frac{\partial L_{\text{Aug}}(\hat{y}_i^{\text{rd}}, \hat{y}_i^*)}{\partial w_i^{\text{map}}} \tag{15}$$

This weight update rule (Equation 15) is also applicable for updating the weight matrix $B_i$ in local error configurations (Equation 7), based on the discrepancy between intermediate readouts $\hat{\xi}_i^t = B_i \xi_i^t$ and the pseudo-targets $\hat{y}_i^*$.

The evaluated CRNN architecture follows the configuration described in Section 4.2 (optimal hyperparameters detailed in Appendix D). Empirical results demonstrate that weighted knowledge distillation yields lower performance compared to networks without additional mapping weights (Table 3). Moreover, this approach incurs a moderate increase in memory consumption due to the inclusion of the extra weight matrices.

## 5  Discussion

The EP framework mimics synaptic learning in the human brain by leveraging biologically plausible local updates. Recent advances have narrowed the performance gap between EP and BP, suggesting its potential as a practical alternative for the training of neural networks. However, current EP frameworks suffer from the vanishing gradient problem, which impedes effective error signal propagation. This study alleviates this limitation by augmenting neuron dynamics with intermediate learning signals, enabling EP to scale up convolutional CRNNs. Experiments on CIFAR-10 and CIFAR-100 datasets with VGG architectures demonstrate state-of-the-art performance without degradation as network depth increases, highlighting the scalability of the augmented EP framework considered here. Compared to BPTT, EP employs spatially and temporally local updates (Ernoult et al., 2020), substantially reducing memory and computational

demands. Its single-circuit architecture and STDP-like weight updates (Scellier & Bengio, 2017) further enhance its suitability for on-chip learning and edge computing, where hardware constraints make efficient local learning especially important. For the KD variant, however, the teaching signals are provided by a separate, already well-trained teacher model on the same task. In this setting, the KD signal can be stored as a static learning signal, while the on-chip learning process remains local and Hebbian, and still preserves the biological plausibility. Moreover, the inclusion of LE can be realized through simple system architecture modifications, making it compatible with hardware-efficient implementations. It would be valuable to explore the generalizability of the proposed framework to more complicated architectures and tasks, including large-scale vision models such as ResNet (He et al., 2016), and transformer-based language models like BERT (Devlin et al., 2019) and GPT-2 (Radford et al., 2019). Finally, we note that during the development of this work, Elayedam & Srinivasan (2025) independently explored EP scalability through residual connections. We view our intermediate signaling approach and their architectural modifications as highly complementary. Combining these independent findings represents an exciting avenue for future neuromorphic research.

## Acknowledgments

This material is based upon work supported by the U.S. National Science Foundation under award No. CAREER #2337646.

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

## A  Vanishing Gradient Problem

In this section, Figure 5 demonstrates the neuron states within a VGG-13 trained by both the standard EP and augmented EP frameworks. In the standard EP setting, the neuron activations gradually decay to zero over epochs, resulting in the vanishing gradient problem. With the augmented intermediate signals, the neuron states remain stable, facilitating information flow from forward-propagating inputs and backward-propagating error signals. In the standard EP framework, weight gradients vanish (Figure 5b). This occurs because the initial layers directly receive the static input data, meaning their activations and gradients are primarily driven by forward propagation. Conversely, the final layers directly receive the injected error signal (the nudge term) during the second phase, resulting in state perturbations. However, both the forward input signals and the backward error signals reduce as they propagate across multiple layers in deep architectures. As a result, intermediate neurons receive insufficient perturbation from either direction, leading to a U-shaped distribution of gradient information across layers. In contrast, when trained using the augmented EP framework, weight gradients maintain a positive magnitude throughout the network (Figure 5d).

The primary driver of the vanishing-gradient problem in deep EP models is the decay of signals across layers. To rule out alternative explanations, we note that proper initialization of neuron states and weights alone is insufficient to mitigate this issue. While we employ random initialization for neuron states and Kaiming uniform initialization for weights in our models, the standard EP framework still failed to converge on deeper architectures without our proposed augmentations.

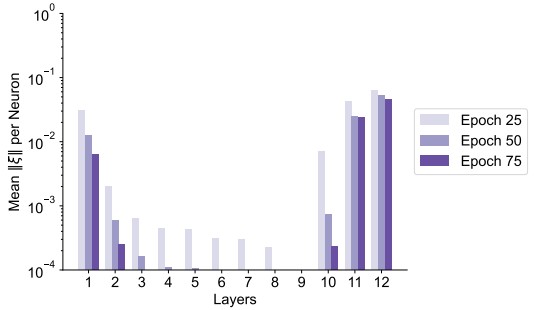

(a) Neuron activations in the standard EP framework.

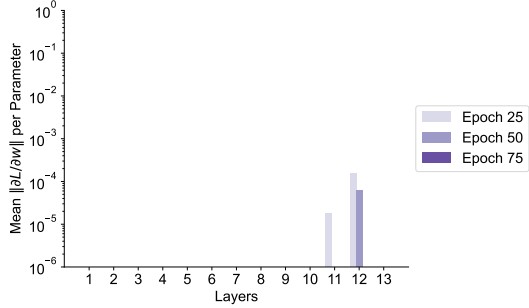

(b) Weight gradients in the standard EP framework.

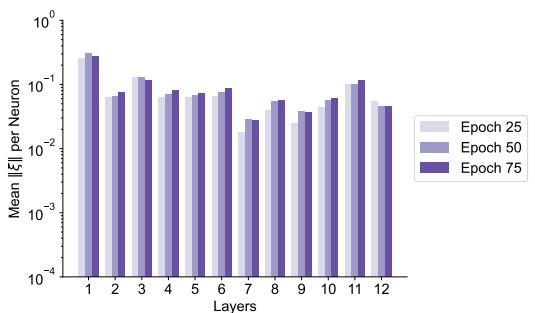

(c) Neuron activations in the augmented EP framework.

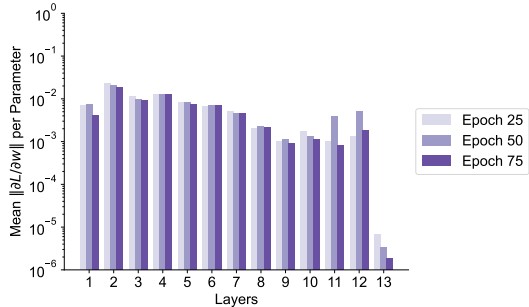

(d) Weight gradients in the augmented EP framework.

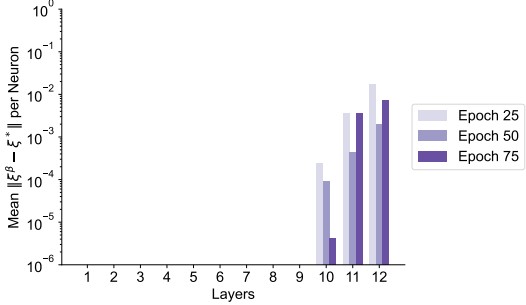

(e) The norm of the difference between neuron activations with and without nudging in the standard EP framework.

Figure 5: The neuron activations and weight gradients at each layer of a VGG-13 network in the nudge phase, trained utilizing standard and augmented EP framework, have been reported for epochs 25, 50, and 75 on the CIFAR-100 dataset. The findings are presented as the mean absolute value of neuron activations per neuron and the mean absolute value of weight gradients per parameter. In 5e, we present a direct comparison of the saturated neuron states at the end of the free and nudged phases in the standard EP framework, demonstrating that neuron activations differ only minimally between the two phases.

Although these initialization methods help standard EP models avoid zero activations at the beginning of training, the activation magnitudes still decay across layers initially. The corresponding weight gradients are small (often falling below $10^{-8}$ in most layers). Over subsequent epochs, the neuron states decay due to a combination of optimizer weight decay and these small initial gradients.

Second, we evaluated whether insufficient nudge phase was responsible. With extensive simulation times, $T_{\text{free}} = 1000$ and $T_{\text{nudge}} = 400$, the standard EP framework failed to converge as depth increased. This empirical evidence suggests that both initialization and extending the simulation time is insufficient to

Table 4: Architectural configurations of convolutional CRNNs based on VGG variants, used in experiments on CIFAR-10 and CIFAR-100 with the EP framework. Each convolutional layer uses a $3 \times 3$ kernel with stride 1 and padding 1. For CIFAR-100, two linear layers are used in LE configuration, while one is used in KD configuration.

| VGG-7 | VGG-10 (11) | VGG-12 (13) |
|---|---|---|
| 7 layers | 10 (11) layers | 12 (13) layers |
| input (RGB image) | | |
| conv3-64 | conv3-64 | conv3-64 |
| | | conv3-64 |
| maxpool | | |
| conv3-128 | conv3-128 | conv3-128 |
| | | conv3-128 |
| maxpool | | |
| conv3-256 | conv3-256 | conv3-256 |
| | conv3-256 | conv3-256 |
| maxpool | | |
| conv3-512 | conv3-512 | conv3-512 |
| | conv3-512 | conv3-512 |
| maxpool | | |
| conv3-512 | conv3-512 | conv3-512 |
| | conv3-512 | conv3-512 |
| maxpool | | |
| FC-512 ($\times 2$) | | |
| FC-OUT[a] | | |

[a] OUT represents the number of classes in the dataset.

overcome the convergence challenges inherent to deep EP architectures. Notably, in our experiments, we observed the vanishing gradient problem when scaling the architecture to depth of 10 and beyond.

## B Datasets Details

We evaluate our models on the CIFAR-10 and CIFAR-100 datasets (Krizhevsky et al., 2009), both comprising 60,000 color images of size 32-by-32 pixels. Each dataset is split into 50,000 training and 10,000 test images. CIFAR-10 consists of 10 classes with 6,000 images per class, while CIFAR-100 contains 100 classes with 600 images per class, making it a more complicated classification task.

## C Architectures

This section delineates the topological configurations of CRNNs employed in the experiments in Table 4.

## D Hyperparameters

This section outlines the hyperparameter configurations utilized in the experiment. Table 5 outlines the hyperparameters for the CIFAR-10 dataset and Table 6 outlines the hyperparameters for the CIFAR-100 dataset. Within both the LE and KD frameworks, the temperature, $\tau$, which controls the smoothness of the output distributions, is set to a value of 4 throughout all experiments. Neuron states of VGG-7-LE, VGG-11-LE, and VGG-13-LE are initialized with zeros, while neuron states of VGG-7-KD, VGG-7-KDW, VGG-10-KD, and VGG-12-KD are initialized with random numbers from a uniform distribution between 0 and 1. The weights of VGG-11-LE, VGG-13-LE, and VGG-12-KD are initialized using Kaiming Uniform Initialization scaled by a factor of 0.5. In all experiments, the hard sigmoid function is employed as the activation

Table 5: Hyperparameters for the CIFAR-10 dataset.

| Hyperparameters | VGG-7-LE | VGG-7-KD | VGG-7-KDW |
|---|---|---|---|
| $\beta$ | 0.25 | 0.15 | 0.5 |
| $T_{\text{free}}$ | 250 | 250 | 100 |
| $T_{\text{nudge}}$ | 50 | 60 | 50 |
| Learning Rate* | $[0.03 \times 7]$ | $[0.03 \times 7]$ | $[0.03 \times 7]$ |
| $\kappa$ | 0.65 | 0.65 | 0.15 |
| Batch Size | 128 | 64 | 128 |
| Epochs | 250 | 250 | 250 |
| Weight decay | 3e-4 | 3e-4 | 3e-4 |
| Momentum | 0.9 | 0.9 | 0.9 |
| $\Upsilon$ | [2,4] | [2,4] | [2,4] |

*A layer-wise learning rate is implemented with $LR \times N$ indicating that the next $N$ number of layers uses learning rate $LR$.

Table 6: Hyperparameters for the CIFAR-100 dataset.

| Hyperparameters | VGG-11-LE | VGG-13-LE | VGG-10-KD | VGG-12-KD |
|---|---|---|---|---|
| $\beta$ | 1.0 | 1.0 | 0.5 | 1.0 |
| $T_{\text{free}}$ | 400 | 140 | 130 | 100 |
| $T_{\text{nudge}}$ | 40 | 50 | 50 | 15 |
| Learning Rate* | $[0.03 \times 1,$ $0.05 \times 7,$ $0.025 \times 3]$ | $[0.03 \times 2,$ $0.05 \times 8,$ $0.025 \times 3]$ | $[0.03 \times 2,$ $0.05 \times 6,$ $0.025 \times 2]$ | $[0.03 \times 2,$ $0.05 \times 8,$ $0.025 \times 2]$ |
| $\kappa$ | 0.85 | 0.85 | 0.85 | 0.65 |
| Batch Size | 128 | 128 | 128 | 256 |
| Epochs | 250 | 250 | 250 | 250 |
| Weight decay | 3e-4 | 3e-4 | 3e-4 | 3e-4 |
| Momentum | 0.9 | 0.9 | 0.9 | 0.9 |
| $\Upsilon$ | [1,3,5,7] | [1,3,5,7] | [0,1,3,5,7] | [1,3,5,7,9] |

*A layer-wise learning rate is implemented with $LR \times N$ indicating that the next $N$ number of layers uses learning rate $LR$.

function, except for the VGG-12-KD configuration, in which ReLU is utilized in the convolutional layers. Random seeds are not manually set; the experiments rely on PyTorch's default handling of stochasticity. All experiments are conducted on a Linux operating system. For training VGG-7-KD and VGG-7-KDW on the CIFAR-10 dataset, the teacher model is a VGG-19 trained to 93.37% accuracy. For VGG-10-KD on the CIFAR-100 dataset, the teacher is a VGG-16 trained to 73.69% accuracy, and for VGG-12-KD on CIFAR-100, a VGG-15 trained to 70.96% accuracy serves as the teacher model. The intermediate logits of the teacher models are extracted from the last layer of each convolutional block. In this study, we adopt the cosine annealing scheduler for the learning rate across all experiments and for intermediate signaling in the majority of cases. The only exception is VGG-12-KD, which utilizes a linear scheduler.

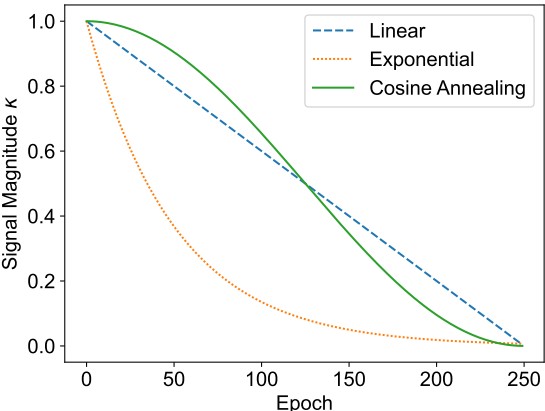

Figure 6: Magnitude decay of various learning signal schedulers evaluated in this study. Here, the parameter $\kappa_{\text{init}}$, $\kappa_{\text{min}}$, and $\gamma$ are set to 1, 0, and 0.02, respectively. The parameter $\mathcal{E}_{\text{total}}$ is designated as 250, yielding empirically optimal performance in Section 4.2.

## E  Learning Signal Scheduler

In this section, we provide the magnitude decay of various learning signal schedulers in Figure 6. We define each scheduler as a function of epoch $\mathcal{E}$ over the total number of epochs $\mathcal{E}_{\text{total}}$.

$$
\begin{aligned}
\kappa_{\text{lin}} &= -\kappa_{\text{init}} \frac{\mathcal{E}}{\mathcal{E}_{\text{total}}} + \kappa_{\text{init}} \\
\kappa_{\text{exp}} &= \kappa_{\text{init}} \cdot e^{-\gamma \mathcal{E}} \\
\kappa_{\text{cos}} &= \kappa_{\text{min}} + 0.5 \cdot (\kappa_{\text{init}} - \kappa_{\text{min}}) \cdot (1 + \cos(\pi \cdot \mathcal{E}/\mathcal{E}_{\text{total}}))
\end{aligned}
\tag{16}
$$

Here, $\kappa_{\text{lin}}$, $\kappa_{\text{exp}}$, and $\kappa_{\text{cos}}$ denote the learning signal magnitudes $\kappa$ corresponding to the linear scheduler, exponential scheduler, and cosine annealing scheduler, respectively. The initial and minimum learning signal magnitudes are indicated by $\kappa_{\text{init}}$ and $\kappa_{\text{min}}$, respectively. The variable $\gamma$ captures the decay ratio in the exponential scheduler.

## F  Formal Mathematical Definitions for Scalar Primitive Function

This section provides the explicit mathematical definitions for the operations comprising the scalar primitive function $\Phi(x, \xi^t, w)$ (Equation 1). The dynamics of convolutional CRNNs are defined using the scalar primitive function, which incorporates convolution, pooling, and generalized scalar products.

**Generalized Scalar Product ($\circ$)**   The generalized scalar product denotes the inner product between two tensors of identical dimensionality. For two arbitrary tensors $A$ and $B$, the operation is defined as the sum of their element-wise products ($i, j$ are indices of matrix):

$$
A \circ B = \sum_{i,j,\ldots} A_{i,j,\ldots} \; B_{i,j,\ldots}
\tag{17}
$$

**Convolution Operator ($*$)**   The convolution operator applies a set of learnable spatial filters to the input feature maps or neuron states. For a given input tensor $x$ with $C_{in}$ channels and a weight tensor $w$ mapping to $C_{out}$ channels, the spatial operation evaluated at output channel $C_{out}$ and spatial coordinates $(i, j)$ with a square kernel of size $k$ and a stride of $s$ is defined as:

$$
(x * w)_{c_{out},i,j} = \sum_{c_{in}=0}^{C_{in}-1} \sum_{m=0}^{k-1} \sum_{n=0}^{k-1} x_{c_{in}, s \cdot i + m, s \cdot j + n} \; w_{c_{out},c_{in},m,n}
\tag{18}
$$

Here, $m$ and $n$ iterate over the spatial dimensions of convolutional kernels utilized. $c_{in}$ and $c_{out}$ are the indices of the input and output channels, respectively.

**Pooling Function ($\mathcal{P}(\cdot)$)**   The pooling function $\mathcal{P}(\cdot)$ represents a spatial downsampling operation applied independently to each channel of the feature map. For a general max-pooling operation with a spatial window size of $k \times k$ and a stride of $s$, the output for a feature map $x$ at channel $c$ and spatial coordinates $(i, j)$ is formally defined as:

$$\mathcal{P}(x)_{c,i,j} = \max_{m \in \{0,\dots,k-1\}, n \in \{0,\dots,k-1\}} x_{c, s \cdot i + m, s \cdot j + n} \tag{19}$$

# G   Proof of Convergence for Augmented EP Framework

In this section, we present a theoretical proof to demonstrate that the convergence guarantees remain valid with the introduction of intermediate learning signals. To clarify the scope of the result, we emphasize that Theorem 3.1 provides a sufficient condition for convergence of the discrete-time augmented EP dynamics. Specifically, we assume that the base transition function is contractive (Lipschitz continuous with constant $K_1 < 1$ in Equation 21). This assumption of a contractive base transition function is consistent with prior EP works (Scellier & Bengio, 2017; Ernoult et al., 2019), which implicitly assume convergence of both the free and nudged dynamics to steady states. In this work, we state this requirement explicitly.

*Proof.* To demonstrate the convergence of the neuron dynamics in our augmented EP framework, we verify that the gradient of the augmented scalar primitive function with respect to the neuron states, $\frac{\partial \Phi_{\mathrm{Aug}}(x, \xi^t, w)}{\partial \xi}$, is Lipschitz continuous. Let $\Phi(x, \xi^t, w)$ denote the base scalar primitive function defined in Equation 1, and $\Phi_{\mathrm{Aug}}(x, \xi^t, w)$ represent the augmented scalar primitive function used during the nudge phase (Equation 10). The corresponding transition functions are defined as:

$$F(\xi^t) = \frac{\partial \Phi(x, \xi^t, w)}{\partial \xi}, \quad F_{\mathrm{Aug}}(\xi^t) = \frac{\partial \Phi_{\mathrm{Aug}}(x, \xi^t, w)}{\partial \xi} \tag{20}$$

Here, $L_{\mathrm{EP}}$ and $L_{\mathrm{Aug}}$ represent the loss functions at the output layer and the intermediate layers chosen for additional signaling, respectively.

Following prior works (Scarselli et al., 2008; Ernoult et al., 2019), we adopt the assumption that the transition function $F(\xi^t)$ is Lipschitz continuous. Under this condition, there exists a constant $K_1$ such that, for any pair of neuron states $\xi_1$ and $\xi_2$,

$$\|F(\xi_1) - F(\xi_2)\| \leq K_1 \|\xi_1 - \xi_2\| \tag{21}$$

In addition, we assume that the loss functions $L_{\mathrm{EP}}$ and $L_{\mathrm{Aug}}$ are smooth and Lipschitz continuous. (The mean squared error (MSE) and cross-entropy loss functions are Lipschitz continuous under standard assumptions of bounded inputs and softmax outputs (Khromov & Singh, 2023; Mao et al., 2023).) Let $K_2$ and $K_3$ denote their respective Lipschitz constants.

According to Equation 10, the transition function of the augmented EP framework is given by:

$$F_{\mathrm{Aug}}(\xi^t) = F(\xi^t) - \beta \frac{\partial L_{\mathrm{EP}}(\xi_{\mathrm{out}}^t, \hat{y})}{\partial \xi} - \beta \kappa \sum_{i \in \Upsilon} \frac{\partial L_{\mathrm{Aug}}(\xi_i^t, \hat{y}_i^*)}{\partial \xi} \tag{22}$$

Using the sum rule for Lipschitz continuity (Rockafellar & Wets, 2009), we conclude that, for any arbitrary neuron states $\xi_1$ and $\xi_2$:

$$\|F_{\mathrm{Aug}}(\xi_1) - F_{\mathrm{Aug}}(\xi_2)\| \leq (K_1 + |\beta|K_2 + |\beta\kappa|K_3) \|\xi_1 - \xi_2\| \tag{23}$$

By Banach's fixed-point theorem (Khamsi & Kirk, 2011), if the base transition map $F$ is Lipschitz continuous with constant $K_1 < 1$, and $L_{\mathrm{EP}}$ and $L_{\mathrm{Aug}}$ are Lipschitz continuous with constants $K_2$ and $K_3$, respectively, then for some choice of hyperparameters $\beta$ and $\kappa$ satisfying $K_1 + |\beta|K_2 + |\beta\kappa|K_3 < 1$, the augmented map $F_{\mathrm{Aug}}$ is a contraction. Therefore, the the discrete-time dynamics converge to the unique fixed point of $F_{\mathrm{Aug}}$. $\qquad\square$

