# OpenReview forum: "Scalable Equilibrium Propagation via Intermediate Error Signals for Deep Convolutional CRNNs"
_TMLR — Accepted by TMLR_

### Review · Reviewer_RUEz · 2026-02-13

**Summary Of Contributions:**

The paper proposes adding intermediary teaching signals to enable the equilibrium propagation algorithm to train deeper convolutional networks.

Strengths:
- state-of-the-art results for equilibrium propagation
Weaknesses:
- the presentation is unnecessarily complicated
- the analysis is sometimes relatively shallow
- the paper barely shifts the needle regarding what is possible with equilibrium propagation, which makes it unlikely to renew the machine learning community's interest in alternative learning paradigms

**Audience:**

Yes

**Audience Explanation:**

A small community at the intersection of physics-based / neuromorphic computing and machine learning is interested in scaling up equilibrium propagation, and a subset of it is likely to be in TMLR's audience.

**Claims And Evidence:**

No

**Claims Explanation:**

While the numbers reported look reasonable, the story around why they work is more flimsy. See "Requested changes" for more details.

**Requested Changes:**

Major issues:
- the authors keep claiming that adding auxiliary teaching signals solves the vanishing gradient problem in the network. There are (at least) 2 possible reasons to have vanishing gradients: initialization issues (as in feedforward networks), or relaxation phases that are too short (so that teaching signals don't have time to be backpropagated to first layers). It is unclear what is the problem in the settings the authors are considering, and thus why auxiliary teaching signals are useful. Given that this is the main contribution of the paper, this point has to be crystal clear. It should also be explained why the gradient magnitude in early layers is relatively large, which seems to be in contradiction with the vanishing gradient story.
- the back propagation issue that should be compared against is recurrent back propagation (Scellier and Bengio 2019 and Zucchet and Sacramento 2022) and not backpropagation-through-time. While it's correct that equilibrium propagation matches the BPTT gradient, it is misleading (at least in the abstract) to claim that it is doing BPTT without more context. This is because there is no temporal credit assignment in equilibrium propagation. Additionally, recurrent back propagation has a much smaller memory footprint than BPTT, which would make comparison with equilibrium propagation in Table 3 more fair.

Minor issues:
- the theoretical introduction of the algorithm should be made more concise: all the results introduced derive from either equilibrium propagation theorem or from the convergence of fixed point iterations scheme, so rederiving these results has limited interest, especially as this is an empirical paper
- is the teacher in the knowledge distillation a feedforward neural network? If so, this is decently surprising that it works, and would be worth emphasizing better

---

> ### Author Response · Authors · 2026-03-18
> **Response to Reviewer Concerns**
>
> ### Responses to Major Issues
> 1. We thank the reviewer for raising these critical points. To clarify the exact nature of the vanishing gradient problem within the EP framework, we added a discussion in Appendix B. We argue that the vanishing gradient problem in our setting is not caused by initialization issues or insufficiently short nudge phases, but rather by the spatial decrement of signals across deep layers. Additionally, due to the nature of the EP framework, we also discuss why we observed a U-shaped distribution of gradient information.
> 2. We thank the reviewer for this insightful distinction. Although prior work established a connection between EP and RBP, such a connection is grounded in continuous-time energy functions. Our work is based on a discrete-time scalar primitive function, which is established in comparison to BPTT according to prior work (Ernoult et al., 2019). Additionally, we chose to benchmark our framework against BPTT because BPTT has consistently served as the standard baseline for performance and computational comparisons in the prior EP literature (Ernoult et al., 2019; Laborieux et al., 2021; Laborieux & Zenke, 2022). Our goal in Table 2 was to illustrate the practical memory savings of our augmented EP framework relative to this widely adopted standard.
>
> ### Responses to Minor Issues
> 1. We appreciate this feedback. While we included these derivations to ensure accessibility for readers outside the EP subfield, we agree with the need for conciseness. Accordingly, we have moved the theoretical proofs to Appendix G, and we are very open to condensing the text. We would gladly accommodate any specific suggestions the reviewer has for derivations.
> 2. We thank the reviewer for highlighting this aspect of our methodology. The reviewer is exactly right. The teacher models utilized in our KD framework are indeed standard, non-dynamical feedforward neural networks trained via conventional BP. As detailed in Appendix~D, we utilize pre-trained VGG-19, VGG-16, and VGG-15 architectures as our teacher models.
>
> ### References
> Maxence Ernoult, Julie Grollier, Damien Querlioz, Yoshua Bengio, and Benjamin Scellier. Updates of equilibrium prop match gradients of backprop through time in an rnn with static input. Advances in neural information processing systems, 32, 2019.
>
> Axel Laborieux and Friedemann Zenke. Holomorphic equilibrium propagation computes exact gradients through finite size oscillations. Advances in neural information processing systems, 35:12950–12963, 2022.
>
> Axel Laborieux, Maxence Ernoult, Benjamin Scellier, Yoshua Bengio, Julie Grollier, and Damien Querlioz. Scaling equilibrium propagation to deep convnets by drastically reducing its gradient estimator bias. Frontiers in neuroscience, 15:633674, 2021.

---

### Review · Reviewer_AWf3 · 2026-03-04

**Summary Of Contributions:**

The authors propose an approach that scales Equilibrium Propagation (EP) to deeper VGG-style CRNNs by adding two learning signals: (i) Local Error (LE): auxiliary readouts with pseudo-targets (in experiments, the true label at each layer) via random projections \(B_i\); (ii) Knowledge Distillation (KD):  matching intermediate representations to a pretrained teacher. The authors give a convergence argument and an augmented gradient formula. On CIFAR-10/100, the method improves over prior EP, with gradient/activation plots showing less vanishing gradient.


**Strengths:**
- The motivation for addressing depth-related gradient decay in EP is clear, and the link to both energy minimization and gradient computation is well explained.
- The intermediate-signal design (LE and KD) is simple and practical
- The augmented gradient and the derivation is easy to follow.
- The empirical gains and the memory comparison vs BPTT (Table 2) are clearly reported. The gradient/activation plots (Fig. 5, Fig. 3) support the claim that vanishing gradients are mitigated.

**Weaknesses:**
- Main experiments are quite limited and use only CIFAR-10, CIFAR-100, and VGG-style CRNNs. Could the authors briefly state the intended scope?
- Maybe this needs to be clarified better: Prior-work VGG-5 and the proposed VGG-7 use different architectures. Without a same-architecture baseline (e.g. VGG-7 without LE/KD), gains could be from depth rather than the method. Could the authors add such a baseline or briefly discuss this?
- When the teacher and student have different depths or channel counts, how are intermediate features or “logits” aligned? A short specification could help in reproducibility.

**Additional Comments:**

N/A

**Audience:**

Yes

**Audience Explanation:**

Yes. Researchers in biologically plausible learning, energy-based models, and neuromorphic computing would find the scaling of EP to deeper conv nets and the LE/KD design relevant. The results and memory comparison vs BPTT are useful.

**Broader Impact Concerns:**

No major ethical concerns.

**Claims And Evidence:**

Yes

**Claims Explanation:**

The claim that intermediate signals mitigate depth-related degradation is supported by better CIFAR results and gradient/activation plots. So while it's limited, the empirical findings support the main claim.

**Requested Changes:**

- Please address the weaknesses mentioned in the Summary section.

---

> ### Author Response · Authors · 2026-03-18
> **Response to Reviewer Concerns**
>
> ### Responses to Weaknesses
> * We thank the reviewer for this question. The intended scope of this work is to demonstrate that EP can scale beyond the shallow architectures, typically limited to five layers in most prior works. By evaluating our augmented framework on CIFAR-10, CIFAR-100, and deep VGG architectures, we establish a rigorous, controlled baseline to prove that intermediate signaling overcomes the vanishing gradient problem. While residual architectures have recently been explored in the EP literature (Elayedam & Srinivasan, 2025), these models are currently limited to 9 layers. In contrast, our method successfully trains architectures up to 13 layers. Since our approach does not rely on residual connections, this result highlights its ability to mitigate vanishing gradients directly. Moreover, our intermediate signaling framework could be naturally integrated into EP-based ResNets in future work to further improve depth and scalability.
> * We appreciate the reviewer highlighting this point.
> We compare a VGG-5 trained with our LE and KD frameworks directly against a standard vanilla VGG-5 on CIFAR-10 dataset. Our methods demonstrate performance improvements, with VGG-5-LE achieving 89.56\% testing accuracy and VGG-5-KD achieving 90.10\%. We highlight our primary contribution regarding the scalability of the proposed framework on the CIFAR-100 dataset, where the standard EP framework fails to converge on deep VGG architectures, for instance with a depth of 10 layers or more.
> * We thank the reviewer for requesting this important implementation detail. In our framework, when the teacher and student have different dimensionality at the intermediate layers, we align the feature dimensions using a learnable linear projection layer before computing the distillation loss. This approach, which we denote as weighted knowledge distillation (KDW), is discussed in the second paragraph of Section 4.3.
>
> ### References
> Sankar Vinayak Elayedam and Gopalakrishnan Srinivasan. Scaling equilibrium propagation to deeper neural network architectures. arXiv preprint arXiv:2509.26003, 2025.

---

### Review · Reviewer_zdip · 2026-03-05

**Summary Of Contributions:**

The paper tackles the issue of the vanishing gradient problem of Equilibrium Propagation (EP) in deep networks. The paper mitigates this by two approaches that incorporate additional signals: Local Error (LE) and Knowledge Distillation (KD). The efficacy of the two methods is empirically verified on VGG networks using CIFAR10 and CIFAR100.

**Strengths**
- Clear motivation

**Weaknesses**
- Missing discussions related to https://arxiv.org/abs/2508.11659, which also reports and tackles vanishing gradient problem?
- Overall, many notations and definitions are used without proper definitions:
   - Definition of the pooling function $\mathcal{P}$, convolution operator $*$, generalized scalar product $\odot$?
   - Definition of the "gradient updates computed by the EP/BPTT algorithms" $\nabla^{\mathrm{EP}}$ and $\nabla^{\mathrm{BPTT}}$
- The main Theorem 3.1 statement is missing two crucial assumptions: transition function $F(\xi^t)$ is Lipschitz, and that $F_{\mathrm{Aug}}$ is Lipschitz with constant **less than $1$**. The latter assumption is very implicit in the proof, which is misleading. The main statement should make these assumptions absolutely clear.
    - IMHO, the proof of Theorem 3.1 itself does not present any new techniques or values. Maybe consider moving this to the Appendix for smoother organization
    - Justifications for the two assumptions are absent. So.. because the transition function is Lipschitz, this would mean that the primitive function $\Phi$ is smooth. Is this assumption realistic or commonly satisfied? (At first thought, this seems to rule out any neural networks containing nonsmooth activations?)
- I think this statement in Section 2.2 is wrong: "a stable state $\xi^*$ that minimizes the scalar primitive function. All that one could say is that the stable state (solution to the fixed-point iteration) is a critical point of the energy functional $E(\xi) := \frac{1}{2} \xi^2 - \Phi(\xi)$. This is also mentioned explicitly in Appendix B.2 of Ernoult et al. (2019).
- Eq. (6) seems to be missing a $-$ sign, according to Theorem 1 of Ernoult et al. (2019)?
- To my understanding, one of the reasons why the EP framework has received so much attention is due to its biological plausibility. But then, I'm confused about whether we can view EP+KD as biologically plausible. In this context, I understand KD as simply a technical fix (rather direct, as it is essentially inserting external teacher signals to all layers of the student) to the vanishing gradient problem. Thus, I'm confused about the meaning of EP+KD, given that there are many, many other machine learning methodologies that are not biologically plausible but have much better performances. From this perspective, I'm not sure whether comparing with EP+KD is a fair comparison at all.
- Similarly, I don't see how EP+LE is biologically plausible and/or suitable for on-chip learning.
- Overall, the novelty of this work is unclear. Because LE and KD are well-established, simply combining them with EP feels incremental and does not yield meaningful new insights.

**Additional Comments:**

This is my first exposure to biologically plausible learning rules such as EP, and thus I may have missed some important parts.

**Audience:**

Yes

**Audience Explanation:**

The topic of biologically inspired (local) learning rules is of great interest to the intersection of neuroscience and machine learning communities.

**Broader Impact Concerns:**

None.

**Claims And Evidence:**

Yes

**Claims Explanation:**

Partially. The theoretical discussions, taken from prior works, have some typos and errors that undermine the accuracy; see weaknesses above. Experimental results support the claimed efficacy of the augmented EP methods.

**Requested Changes:**

**aesthetic changes**
- Some equations should be made into a single line, e.g., Eq. (1), (13), (14), (15), (16), unless the authors meant to emphasize certain part, in which the paper should make it clear.
- "Theoretical support for Theorem ?? is established in ??" should be changed to explicitly referring to the precise theorem number of the corresponding reference. I would suggest using something like \begin{theorem}[Theorem ? of \citet{??}].

**Questions**
- In Section 3.1, the authors state that information concentrated in the initial and final layers is a consequence of the traditional EP framework. But I don't understand why this directly implies that the initial and final layers are the ones gaining all the information. Is this trivial?
- As stated in Theorem 3.1, the condition that $F_{\mathrm{Aug}}$ is contractive is sufficient for the fixed-point iteration (discrete-time neuron dynamics) to converge. Then, for the experiments, do the authors perform some sort of rescaling to satisfy the contractive mapping property? If not, then do the authors empirically observe that the "raw" discrete-time neuron dynamics converge? Can the authors explain why? (Maybe the initial point is already close to a locally contractive neighborhood)
    - A follow-up: if the contractiveness is a sufficient condition that is often not satisfied, then what is the point of Theorem 3.1?
- When the authors mention that SGD with momentum is used, does this mean that the optimizer stays the same, and the only difference is that the usual backprop-based weight gradient is replaced with $\nabla_{\mathrm{Aug}}^{\mathrm{EP}}$?
- Is EP+KD biologically plausible and/or suitable for on-chip learning, given that it requires two models and we need to continually flow the information from the teacher's layers to the student's layers, not to mention the requirement of the teacher? Same question for EP+LE: is this biologically plausible and/or suitable for on-chip learning?

---

> ### Author Response · Authors · 2026-03-18
> **Point-by-Point Response to Reviewer**
>
> ### Responses to Weaknesses
> * We thank the reviewer for directing us to this relevant work. We have updated our manuscript to include a discussion of this work in Section 1 (Skip connection part).
> * We thank the reviewer for highlighting these areas where mathematical definitions were lacking. In the revised manuscript, we have included Appendix F to explicitly define these operations. Furthermore, in Section 2.3, we have explicitly clarified the notations used in Theorem 2.2.
> * We thank the reviewer for this careful observation.
>     * We agree that the original statement of Theorem 3.1 did not make all required assumptions sufficiently explicit. We have revised the statements and proof accordingly so that these assumptions are stated clearly rather than left implicit. We also clarify that these assumptions are consistent with prior literature underlying EP (Ernoult et al., 2019; Laborieux et al., 2021). Our theoretical result serves to show that the introduction of intermediate learning signals remains compatible with this standard fixed-point convergence framework.
>     * Additionally, we agree that the proof of Theorem 3.1 is to show that the proposed augmentation remains compatible with the standard discrete-time fixed-point framework used in prior EP analyses. For this reason, we have moved the detailed proof to Appendix G and retained only a brief summary in the main text.
>     * Finally, we clarify that the Lipschitz continuity of the transition function $F$ does not imply that its primitive $\Phi$ is smooth. Prior EP works only assume the existence of the first- and second-order derivatives of $\Phi$ with respect to $\xi$, rather than strict smoothness (Ernoult et al., 2019).
> * We sincerely thank the reviewer for their careful reading and for catching this statement inaccuracy. We have corrected this statement in Section 2.2 of the revised manuscript to accurately reflect this concern.
> * We thank the reviewer for carefully checking the derivations and pointing this out. We have corrected Equation (6) in the revised manuscript to include the missing minus sign.
> * We thank the reviewer for raising this important question. Regarding KD, we agree the pre-trained teacher introduces an artificial element at the system level. However, the weight updates within the student network remain entirely local and Hebbian, preserving EP's computational mechanism. We demonstrate how an external signal can guide learning.  Because our target application is edge computing, where the primary bottleneck is energy-efficient, local learning on the device, storing a static teacher to enable on-chip student learning is a practical tradeoff. Regarding LE, this LE approach is fully BP-independent and fundamentally biologically plausible, relying solely on local information to successfully scale deep architectures. We have added a dedicated discussion of these hardware and bio-plausibility considerations in Section 5.
> * We thank the reviewer for their feedback, but respectfully emphasize that integrating LE and KD into the EP framework is highly non-trivial and extends far beyond a simple incremental combination. Unlike standard BP-trained models, EP operates on a delicate dynamical system governed by a scalar primitive function $\Phi(x,\xi^{t},w)$. Improperly injecting auxiliary teaching signals into this process risks disrupting the energy landscape, preventing convergence, and causing gradient computations to collapse entirely. Our primary contribution and novelty lie in providing theoretical guarantees for the safe embedding of auxiliary signals, and empirically verifying the resulting performance gains. By safely embedding these signals, we overcome the vanishing gradient bottleneck, enabling the successful training of significantly deeper EP models than prior works.
>
> ### References
> Maxence Ernoult, Julie Grollier, Damien Querlioz, Yoshua Bengio, and Benjamin Scellier. Updates of equilibrium prop match gradients of backprop through time in an rnn with static input. Advances in neural information processing systems, 32, 2019.
>
> Axel Laborieux, Maxence Ernoult, Benjamin Scellier, Yoshua Bengio, Julie Grollier, and Damien Querlioz. Scaling equilibrium propagation to deep convnets by drastically reducing its gradient estimator bias. Frontiers in neuroscience, 15:633674, 2021.

---

> > ### Author Response · Authors · 2026-03-18
> > **Point-by-Point Response to Reviewer (Part 2)**
> >
> > ### Responses to Aesthetic Changes
> > We thank the reviewer for pointing this out. We consolidated these equations to improve the overall flow and readability of the mathematical derivations in the revision. We also adopted the suggested formatting for Theorem 2.1.
> >
> > ### Responses to Questions
> > * We thank the reviewer for pointing this out. To clarify this point, we added a discussion in Appendix B. Specifically, the concentration of information in the initial and final layers is a direct physical consequence of how signals are injected into the EP system. The initial layers directly receive the static input data, meaning their activations and gradients are strongly driven by the forward propagation. Conversely, the final layers directly receive the injected error signal (the nudge term) during the second phase, resulting in large state perturbations. Because both the forward input signals and the backward error signals physically decrease as they propagate across multiple layers in a deep architecture, the intermediate neurons fail to receive sufficient perturbation from either end. This creates a U-shaped distribution of gradient information (see Figure 5).
> > * We thank the reviewer for this insightful theoretical question As briefly noted in Appendix D, the weights for our deeper models are initialized using Kaiming Uniform Initialization scaled by a factor of 0.5. Furthermore, as detailed in our hyperparameter tables, the nudging factor $\beta$ and the auxiliary signal magnitude $\kappa$ are bounded to be less than or equal to 1. Under these specific tuning choices, we empirically observe that the discrete-time neuron dynamics successfully converge. The role of Theorem 3.1 is to guarantee that the inclusion of intermediate learning signals does not break the convergence guarantees assumed in prior works.
> > * We thank the reviewer for the question. This is exactly correct. In our implementation, the standard PyTorch SGD optimizer, including its momentum and weight decay mechanics, remains completely unchanged. The only difference from a deep learning pipeline is how the gradients are computed.
> > * We thank the reviewer for raising this important question. Regarding KD, we agree the pre-trained teacher introduces an artificial element at the system level. However, the weight updates within the student network remain entirely local and Hebbian, preserving EP's computational mechanism. We demonstrate how an external signal can guide learning.  Because our target application is edge computing, where the primary bottleneck is energy-efficient, local learning on the device, storing a static teacher to enable on-chip student learning is a practical tradeoff. Regarding LE, this LE approach is fully BP-independent and fundamentally biologically plausible, relying solely on local information to successfully scale deep architectures. We have added a dedicated discussion of these hardware and bio-plausibility considerations in Section 5.

---

> > > ### Comment · Reviewer_zdip · 2026-03-25
> > >
> > > I thank the authors for their detailed responses.
> > >
> > > Regarding the significance of this work as well as the empirical parts, as I do not have sufficient domain knowledge beyond my original reviews, I will leave those points to the AE and other reviewers.
> > >
> > > There are still several mathematical inconsistencies that I would like to point out:
> > > - In Section 2.2, the revision states that "critical point of the primitive function". This is indeed true for continuous-time dynamics, under certain assumptions for convergence (e.g., Łojasiewicz inequality). But in discrete-time (which is the sole focus of this paper to my understanding), this is not the case. One should say that the recursion, if convergent, leads to $\xi^* = \nabla Phi(\xi^*)$, and so the authors should say "fixed point of the gradient field of the primitive function $\nabla \Phi$" or "critical point of the energy functional $E = \frac{1}{2} \xi^2 - \Phi(\xi)$".
> > > - I agree with reviewer RUEz that this is a mainly empirical paper, as most of the theorems are from prior works, and the "theoretical novelty" (Theorem 3.1) is also, IMHO, direct corollaries of prior works combined with appropriate Lipschitzness assumptions.
> > >    - Theorem 3.1 is still missing all the assumptions used to prove the convergence.
> > >    - When I said "smooth" in my original review, I meant $L$-smooth as in optimization theory (apologies for the confusion here). So then, if the transition functions $F$ (Eqn. 20, derivatives of the primitive function $\Phi$) are $K_1$-Lipschitz, then *by definition*, $\Phi$ is $K_1$-smooth. But again, I don't see the value of the theoretical statements here.
> > > - The authors should clarify what they mean by "Our primary contribution and novelty lie in providing theoretical guarantees for the safe embedding of auxiliary signals, and empirically verifying the resulting performance gains." in the rebuttal. How does the theoretical guarantees have any connection with the performance gains? To claim this, I believe that the theoretical results should include something like the proposed framework provably avoids vanishing gradient issues.
> > >
> > >
> > > Overall, my opinion is that *solely* from a theoretical viewpoint, this paper does not add any new value.

---

> > > > ### Author Response · Authors · 2026-03-28
> > > > **Point-by-Point Response to Reviewer**
> > > >
> > > > * We sincerely thank the reviewer for pointing out this precise mathematical distinction. We have updated the terminology in Section 2.2 to be mathematically rigorous for our discrete-time framework.
> > > > * We thank the reviewer for the clarification. We agree that assuming the transition function is Lipschitz strictly implies the primitive function is smooth, which technically rules out the non-smooth operations (like Hard Sigmoid or Max-Pooling) used in our experiments. However, making this strict assumption is standard practice in EP theory to guarantee convergence. Foundational EP models \citep{scellier2019equivalence} similarly govern continuous-time dynamics via an ODE that strictly requires a smooth energy function for a unique solution. We follow this established precedent. Our theoretical results assume a regime where the transition function is Lipschitz continuous, while our empirical implementations deploy non-smooth variants for practical purpose. Additionally, we have added assumptions to Theorem 3.1 in the revision.
> > > > * We apologize for the confusion caused by our previous phrasing. Our intention was to highlight that our contribution has two distinct parts. One is that we propose an architectural framework (adding LE/KD signals) that empirically mitigates vanishing gradients and improves EP scalability. Another is that Theorem 3.1 simply demonstrates that embedding these auxiliary signals preserves the necessary convergence of the neuron dynamics without breaking the underlying EP system. We agree that a rigorous mathematical proof demonstrating the absence of vanishing gradients would be highly valuable. However, deriving such strict bounds for deep, non-linear architectures is notoriously difficult, placing this specific theoretical guarantee outside the scope of our work. Instead, our contribution lies in empirically demonstrating that our framework successfully circumvents the vanishing gradient problem in practice.

---

### Decision · Action_Editor_GyQf · 2026-04-08

**Recommendation:** Accept with minor revision

**Additional Comments:**

Please perform two minor revisions:

- Formulate more clearly and cautiously, also in the abstract, how the LE+KD method addresses the vanishing gradients in EP and what this means for the applicability of EP. Please also clearly point out that knowledge distillation requires another, already well-trained model on the same task to provide the teaching signal.
- The paper is still unclear about the point in how far a vanishing of forward signals in responsible for the vanishing of backwards gradient information. While it is stated that the initialization is such that activation variance is preserved across layers, this is not seen explicitly, e.g. in Fig.5a, and the initialization width of the weight parameters has apparently not been varied or optimized as part of the hyperparameters. Please clarify and if necessary address: Is the activation signal constant across layers at initialization, and the decay of activations in the non-augmented EP models only due to the weight decay term in the loss?

**Audience:**

Yes

**Audience Explanation:**

The referees agreed that making EP applicable to deeper and larger architectures is relevant and that the paper presents a step in this direction.

**Claims And Evidence:**

Yes

**Claims Explanation:**

While the referees agreed on the overall interest of making equilibrium propagation (EP) applicable to deeper architectures, they gave mixed reviews of the quality of the presented evidence.

The referees and authors agreed that the main novelty of the paper are not the theoretical derivations, but the empirical method of augmenting EP with local targets at intermediate layers.

The experiments restoring gradient signals and trainability in all layers are convincing. The paper does not present an investigation of why EP suffers from quickly vanishing gradients and how the cause of this problem can be fixed, however. Instead, it shows that the missing gradient signals can be replaced by additionally injecting local gradients through 1) direct top-down error feedback to lower layers and 2) local targets provided by knowledge distillation from a pre-trained teacher model.
Both of these methods are valid but by changing the learning task to a local one side-step the problem of EP insufficiently propagating gradient signals through multiple layers. The claim that EP is thereby scalable to deep architectures is therefore too strong and should be worded more carefully.

Another unclear point was in how far the forward propagation of activations is partly responsible the vanishing of backwards gradients in the presented experiments.

---

> ### Author Response · Authors · 2026-05-08
> **Responses to comments**
>
> * We thank the reviewer for this helpful suggestion. We have revised Section 3.2 and the Discussion to clarify both the mechanism and the scope of our claim. In particular, we now state more cautiously that intermediate learning signals mitigate rather than universally solve the vanishing-gradient problem in deep EP-trained CRNNs.
> * We thank the Action Editor for this important question. In our implementation, the weights are initialized using Kaiming uniform initialization, and weight decay is applied during optimization. We do not observe that activations are exactly constant across layers at initialization. The activation decay observed in the non-augmented EP model can be partially influenced by weight decay. We have revised the Appendix A to clarify these points in the camera-ready version.